# The Global Burden of Valvular Heart Disease: From Clinical Epidemiology to Management

**DOI:** 10.3390/jcm12062178

**Published:** 2023-03-10

**Authors:** Gloria Santangelo, Francesca Bursi, Andrea Faggiano, Silvia Moscardelli, Pasquale Simone Simeoli, Marco Guazzi, Roberto Lorusso, Stefano Carugo, Pompilio Faggiano

**Affiliations:** 1Department of Cardio-Thoracic-Vascular Diseases, Foundation IRCCS Ca’ Granda Ospedale Maggiore Policlinico, 20122 Milan, Italy; 2Division of Cardiology, Department of Health Sciences, San Paolo Hospital, University of Milan, 20122 Milan, Italy; 3Department of Clinical Sciences and Community Health, University of Milan, 20122 Milan, Italy; 4Cardiothoracic Surgery Department, Heart and Vascular Centre, Maastricht University Medical Centre (MUMC), The Cardiovascular Research Institute Maastricht (CARIM), 6229 ER Maastricht, The Netherlands; 5Cardiothoracic Department Unit, Fondazione Poliambulanza, Via Leonida Bissolati 57, 25100 Brescia, Italy

**Keywords:** heart valve disease, epidemiology, aortic stenosis, aortic regurgitation, heart valve center, heart team, mitral annulus calcification, mitral annular disjunction

## Abstract

Valvular heart disease is a leading cause of cardiovascular morbidity and mortality and a major contributor of symptoms and functional disability. Knowledge of valvular heart disease epidemiology and a deep comprehension of the geographical and temporal trends are crucial for clinical advances and the formulation of effective health policy for primary and secondary prevention. This review mainly focuses on the epidemiology of primary (organic, related to the valve itself) valvular disease and its management, especially emphasizing the importance of heart valve centers in ensuring the best care of patients through a multidisciplinary team.

## 1. Introduction 

### 1.1. Global Burden of Valvular Heart Disease (VHD)

VHD is a major contributor to the loss of physical disability and worsening quality of life, representing a leading cause of cardiovascular morbidity and mortality worldwide [1]. The comprehension of the geographical and temporal trends and changes in VHD epidemiology are crucial for advances in clinical practice and the development of effective health policy for primary and secondary prevention [2]. Although population-based studies are a suitable methodology for investigating the prevalence of a disease, in the context of VHD, they require complete echocardiographic examinations in a large sample that is well representative of the population. Moreover, they heavily rely on routinely collected data (including ICD-10 codes). This epidemiological approach can be unreliable, as post-mortem analyses have revealed the true prevalence of VHD to be significantly greater than that clinically coded and reported [3]. Indeed, population-based data tend to be collected only when VHD is at least moderate or clinically relevant, with patients referred to a diagnostic test because of symptom complaints or the presence of some clinical indications. Furthermore, the limited access to the VHD diagnostic technique likely results in a significant underreporting of VHD, especially in low- or middle-income/resource-poor countries [4]. Finally, the specific causes of VHD can be misclassified, especially in areas where rheumatic heart disease (RHD) is endemic and the classification of VHD is easily prone to error [5]. RHD remains by far the most common cause of primary VHD worldwide. The most marginalized and poorest populations regionally, nationally, and at a subnational level are not showing signs of improvement, and people continue to die early from RHD. Despite a substantial reduction in the burden of global poverty over the past 40–50 years, the global prevalence of RHD has been rising steadily since 1990, reaching 40.5 million people affected in 2019. Globally, RHD-related deaths decreased until 2012 but have stabilized since then and have even started increasing since 2017 [6]. It is true that RHD is now considered very rare in Western countries, but it is worth noting that, in 2019, there were an estimated 152,700 new cases and 2.3 million people living with RHD across the European Society of Cardiology (ESC) member countries, with a clear gradient related to the income of European countries: RHD incidence was twice as high in middle-income countries compared with high-income ones [7]. Conversely, there is a clear predominance of degenerative VHD (especially aortic and mitral) and infectious endocarditis in high-income countries. The incidence of calcific aortic valve disease (CAVD) has increased sevenfold during the last 30 years, with age-standardized rates four times as high in high-income compared with middle-income countries. Similarly, the absolute prevalence of primary mitral regurgitation (PMR) has increased significantly over the past 20 years (by 70% between 1990 and 2017), as well as the global absolute prevalence of non-rheumatic endocarditis (by 44% since 1990) [8].

### 1.2. Factors Responsible for Changes in the Epidemiology of VHD

Considering the above, the resultant VHD disease burden is only projected to increase in the coming decades, with a consequent worsening of related morbidity and mortality. Among the multiple factors involved in the geographical and temporal trends and changes in VHD epidemiology, the following deserve to be mentioned:-Population aging. Worldwide life-expectancy has improved over time [9]. As populations age, some VHD historically considered “age-related”, such as CAVD and degenerative mitral regurgitation (MR), have time to become symptomatic and, consequently, be detected. Furthermore, major advances in the treatment of VHD have allowed for the improvement of long-term survival, thus increasing global prevalence.-Availability of imaging techniques and accessibility to diagnosis and treatment. Some VHDs, such as RHD, are associated with poverty, inequality, overcrowding, and a lack of healthcare facilities, including access to treatment. Despite the worldwide rates of extreme poverty having fallen in the past 40–50 years, RHD prevalence continues to increase unabated. The continuing increase in prevalence cannot be attributed only to population aging but likely reflects the increased global awareness, the increasing availability of echocardiography for case definitions, the greater access to treatment, and the consequent improved survival in many low-income countries [8]. Similarly, in middle- and high-income countries, the advent of progressively advanced echocardiography machines, the improvement of operator skills, and the shift of healthcare systems towards prevention policies have meant that degenerative VHDs (for example, aortic stenosis—AS—and primary MR) are diagnosed more easily, even if mild and asymptomatic.-Migration flows. The spread of urbanization worldwide during the transition from agricultural to industrial activity and service economies is responsible for exposing increasing numbers of people to traditional and non-traditional cardiovascular risk factors, such as smoking, hypertension, obesity, diabetes, air pollution, and stress, which are strongly associated with VHD, as calcific AS [10]. As fertility rates fall below replacement levels in high-income countries and population aging increases, the need for young immigrant groups to supplement the workforce and provide support for the elderly becomes more pressing [10]. Socioeconomic deprivation is common in immigrant groups and is compounded by a range of health inequalities, many of which are indirect consequences of discrimination and racism [11]. Immigrants are more likely to have RHD, which is usually very rare among high-income country inhabitants, and are exposed to cardiovascular risk factors that make them more prone to develop early cardiovascular disease, including VHD.-The improvement in valve surgery and the advent of transcatheter procedures. Valve repair and replacement are now routine cardiac surgical procedures with increasing safety and durability, and they are responsible for the increased survival of patients with VHD. Even in low- and middle-income countries, the access to heart valve surgery is increasing day by day. Nonetheless, due to the combination of high device costs and available workforce capacity and expertise, large discrepancies still persist. Indeed, while in some West-African countries, just 1 cardiac surgeon per 10 million inhabitants is available [12], in some European Countries (e.g., Finland and Lithuania), this proportion goes up to 1 per 200 [7]. In the last decade, transcatheter interventions for VHD have had a rapid increase, allowing for the successful treatment of VHD in old, fragile, and high-surgical-risk patients, with procedural numbers growing exponentially [13]. Moreover, the intrinsic possibility of performing multiple sequential and “staged” transcatheter procedures sparks a real paradigm change for the management of patients with mixed valve disease, who are no longer strictly obliged to undergo cardiac surgery [14]. Confirming the popularity of transcatheter interventions for VHD, a median of 60.0 transcatheter aortic valve implantation procedures and 5.4 percutaneous mitral valve (MV) repairs (PMVR) per million inhabitants were performed in 2019 in European countries [7]. For example, as shown in Figure 1, a relevant increase in percutaneous aortic and MV interventions occurred in Italy in the last five years.

The main downside related to the large-scale diffusion of VHD surgery and transcatheter intervention is that survivors remain at risk of structural valve deterioration, valve thrombosis, and prosthetic valve endocarditis and frequently require re-intervention [15].

-The increase in the predisposing factors of infective endocarditis. The last 20 years have witnessed an increase in all the main factors predisposing to infective endocarditis, such as an aging population, the increased use of intracardiac, vascular, and valvular devices (e.g., pacemakers, defibrillators, biological and mechanical valve prostheses), epidemic levels of opioid addiction, and associated injection drug use. Furthermore, the inadequate and unholy antibiotic use, with the consequent increase in antibiotic resistance, has led to the dangerous shift from Streptococcus to more virulent microorganisms, such as Staphylococcus and Enterococcus, as more common causative organisms of infective endocarditis [15].

Considering the above, the aim of our review is to broadly discuss the epidemiology of VHD, emphasizing the wide geographical, socio-economic, and demographic differences that characterize the spectrum of the most frequently clinically encountered VHD phenotypes: CAVD, degenerative mitral valvulopathy, RHD, and mixed-valve disease. With the exception of some hints regarding secondary VHD (functional, resulting from the pathology in extra-valve cardiac structures such as ventricles and atria), this review mainly focuses on primary VHD (organic, related to the valve itself), their epidemiology, and related treatments.

## 2. Aortic Stenosis

According to the Euro Heart Survey on Valvular Disease, AS is the most common VHD in developed countries, and its etiology is degenerative-calcific in most patients (81.9%), rheumatic in 11.2%, congenital in 5.6%, and post-endocarditis in the remaining 1.3% [16]. The prevalence of aortic sclerosis in individuals over 75 is about 40% [17], and, as the degenerative process very slow, only 2% progress every year to hemodynamically significant AS [18]. In the Chinese population, calcific-degenerative AS has a prevalence of 21.9% among patients over 70 years old [19]. Rheumatic AS is frequently encountered in Asia, with a reported prevalence of 4.54 in India, 1.86 in China, and 1.3 in Bangladesh per 1000 persons [2]. A non-negligible percentage of patients with AS may have concomitant cardiac amyloidosis, commonly due to wild-type transthyretin [20]. Radiation therapy to the mediastinum can be associated with significant valvular abnormalities (typically involving aortic and mitral valves), usually manifesting as progressive valve thickening and calcification and ultimately resulting in valve restriction and dysfunction, which present as stenosis and/or regurgitation. Often, in addition to the valves, surrounding structures such as the valve annulus, subvalvular apparatus, and aorto-mitral curtain are also frequently involved [21].

Osnabrugge et al. have shown that the prevalence of all types of AS in the elderly is 12.4% and the prevalence of severe stenosis is 3.4% [22]. Bicuspid aortic valve (BAV) shows a faster progression to degeneration and an earlier clinical presentation compared with a trileaflet valve [23]. Obesity and hypercholesterolemia in combination with inflammation at the age of 50 years seem to be associated with the degeneration of the aortic valve [24]. Despite atherosclerotic disease and AS sharing hypercholesterolemia and hypertension as risk factors [25], diabetes seems to predispose to the development of AS and to faster its progression, especially from mild to severe stenosis, but the data supporting this hypothesis are still inconsistent [26]. In the last decade, an association between chronic kidney disease and CAVD has emerged; particularly, the prevalence of AS is higher among dialysis patients compared to the general population (7.8% vs. 3.5%) [27], and it progresses more rapidly [28]. Furthermore, peritoneal dialysis, compared with hemodialysis, might reduce the onset and progression of aortic valve calcification and improve AS symptomatology [29].

The onset of symptoms is the greatest marker of progression, requiring a multidisciplinary evaluation in order to choose the most appropriate modality of intervention [30]. Valvular assessment requires a multimodality imaging approach: transthoracic Doppler echocardiography is the first modality used to identify and quantify valvular stenosis. Figure 2 depicts the echocardiographic characteristics of severe AS. The timing of the follow-up is chosen considering the annual decrease in the valve area, the increase in the peak velocity, and the mean transvalvular gradient. Computed tomography (CT) is considered for the pre-operative evaluation of patients undergoing valvular replacement and for the quantification of the aortic calcium score.

Unfortunately, no medical therapy is able to modify the course of the disease; consequently, the only treatment is surgical aortic valve replacement (SAVR) or transcatheter aortic valve replacement (TAVR). Severe AS is the most common indication for valvular intervention in developed countries [16]. In the 2021 European Guidelines for the management of VHD, the intervention is indicated in cases of severe AS among symptomatic patients (regardless of the left ventricular ejection fraction, LVEF) and among asymptomatic patients with systolic dysfunction (without other causes of the impairment of systolic function) or an exercise test that is positive for the development of symptoms or for a sustained fall in blood pressure (at least 20 mmHg). The choice between SAVR and TAVR intervention must be based on the Heart Team (HT) evaluation of clinical, anatomical, and procedural factors, with TAVR being preferred in patients who are at a higher risk and/or unsuitable for surgery.

Early SAVR seems to be the best strategy for asymptomatic patients with very severe AS (defined by the presence of an aortic valve area of 0.75 cm^2^ or less, associated with a peak aortic jet velocity of at least 4.5 m/s or a mean gradient of at least 50 mmHg) and preserved LV systolic function. According to the RECOVERY trial, this approach allows for a lower incidence of the composite of operative mortality or death from cardiovascular causes among patients with very severe AS compared with the conservative watchful waiting approach [31]. The AVATAR trial showed that early SAVR was beneficial in preventing adverse events among patients with asymptomatic severe AS [32]. Lastly, balloon aortic valvotomy may be considered as a bridge to SAVR or TAVR in hemodynamically unstable patients [33].

## 3. Aortic Regurgitation

Aortic regurgitation (AR) is the fourth VHD in the world [34], but in developed countries, where the rheumatic fever is rare, it reaches the third position in the group of non-rheumatic VHD [35]. The prevalence is variable, probably due to an underdetection of asymptomatic patients, reaching 1.6% among United Kingdom (UK) subjects aged >65 years old [36] and 4.9% among participants in the United States Framingham study [37]. The true prevalence in resource-poor, developing countries is challenging to obtain due to the limited access to echocardiography.

AR can result from aortic valve leaflet disease and/or alterations in the aortic root and ascending aorta, and its development can be slowly progressive or acute. Other causes of acute AR are traumatic rupture, acute aortic dissection, balloon aortic valvuloplasty complications [38], myxomatous valve degeneration, aortitis secondary to syphilis, or giant cell arteritis [39].

The most common causes of AR in Western countries are congenital BAV and the calcific degeneration of leaflets [40]. While men diagnosed with BAV are more likely to develop AS, women principally develop AR, and about 30% of patients with BAV are diagnosed with moderate–severe regurgitation [38]. Hypertension—in particular, diastolic hypertension—is recognized as a risk factor for both aortic root dilatation and valve regurgitation [39,41].

According to Yang et al., the risk factors associated with the progression of regurgitation are the male sex, a younger age at diagnosis, the presence of BAV, a larger effective regurgitant orifice and a greater regurgitant volume, and the dilation of the aortic annulus and sinotubular junction [42].

Different genetic syndromes are associated with the development of AR, such as Marfan syndrome and Turner’s syndrome. In Marfan syndrome, the mutation of the FBN1 gene leads to progressive aortic root dilatation and the consequent development of AR, beyond the increased risk of aortic dissection [43].

Sachdev et al. evaluated, with transthoracic echocardiography, 253 patients with Turner’s syndrome and found a significant correlation with AR; in particular, a high prevalence of BAV emerged. In this study, the majority of patients had trivial AR (55%), 30% had a mild regurgitation, and moderate to severe regurgitation was present in 15% [44].

While there is no medical therapy that is able to modify the progression of AR, in Marfan syndrome and in BAV, the use of losartan associated with atenolol seems to reduce the rate of dilatation of the aortic root [45,46].

## 4. Mitral Stenosis

Despite a global decline in its prevalence, with the majority of patients living in the developing world, the most common etiology of mitral stenosis (MS) remains RHD (79%) [16]. RHD is known to be associated with a low economic status, it being more prevalent in low-income rural areas. For example, in Brazil, there has been a significant reduction in new cases of RHD in large urban centers, but prevalence is still high in rural areas [47]. The same goes for countries such as Southern China and India, where RHD is still the main etiology of mitral valvulopathy [48,49].

The global prevalence of RHD is around 1 per 1000 in children aged 5–14 years, with a different prevalence among regions.

It is important to emphasize that, in the majority of countries, the prevalence of RHD is underestimated because it is calculated on the basis of partial and sectorial observations rather than on the basis of large systematic and comprehensive epidemiological studies.

The clinical presentation of rheumatic MS is variable. In countries with a high disease prevalence, it presents at a young age (teen years to 30 years old) with commissural fusion but pliable noncalcified valve leaflets, often associated with regurgitation. In regions with a low disease prevalence, usually Western countries, it is usually detected in older patients (aged 50 to 70 years) who present calcified fibrotic leaflets in addition to commissural fusion and subvalvular involvement [50].

Transthoracic echocardiography is indicated to evaluate the morphology, establish the diagnosis, and quantify the degree of stenosis. The guidelines suggest repeating echocardiograms at intervals dictated by the valve area, even in asymptomatic patients [1]. Symptomatic patients with severe MS are candidates for percutaneous mitral commissurotomy or, if contraindicated, a surgical one or mitral valve replacement [1]. Other infrequent etiologies are congenital MS, which is typical among newborns and infants, radiation-induced MV disease, which usually occurs two or three decades after the chest radiotherapy [51], and degenerative MS, which is found with increasing frequency among elderly subjects from high-income countries. The hallmark of degenerative MS is mitral annulus calcification (MAC). It is characterized by the presence of dense calcification at the base of the mitral leaflets between the LA and LV. Mayo Clinic researchers reported that the MAC prevalence is about 23% in the general population [52,53]. MAC is associated not only with increased cardiovascular events but also with all-cause death. The occurrence of MAC increases the probability of valvular dysfunction, with consequent implications for the treatment of MV. Indeed, patients with MAC are typically older, with several comorbidities, and, due to the highly calcific MV apparatus, they have extremely difficult anatomies that are not suitable for repair. Therefore, surgery for MVD associated with MAC is often delayed until symptoms are severely limiting, or conservative therapy is selected [54]. In patients with calcified valves, the systemic biomarkers of inflammation are elevated and correlate with mortality; thus, it is possible that they play a role in the increased mortality observed in patients with MAC [55,56]. The copresence of MAC and MVD was observed to importantly increase the mortality [57]. The burden of atherosclerotic risk factors, such as smoking, dyslipidemia, obesity, and high systolic blood pressure, not only increase the risk of the calcification of the MV apparatus but also contribute to CAVD, explaining the frequent coexistence of the two-valve stenosis [5,58].

## 5. Mitral Regurgitation

MR is the second most frequent VHD in Europe [59]. MR is divided into primary (PMR), or organic, and secondary, or functional, which is important for the choice of the therapeutic approach. The common causes of MR are prolapse (22%), rheumatic disease (16%), ischemic disease (30%), and dilated cardiomyopathy (26%) [60]. Although no large epidemiological studies are available, MR is prevalent among young adults in countries with endemic rheumatic fever [61]. Regarding mitral valve prolapse (MVP), it is one of the most frequent VHDs, reaching a prevalence of 2–4% in Western populations, similar to that found in Asian populations [62].

PMR is characterized by a primary lesion of one or more components of the MV apparatus. Degenerative etiology (fibroelastic deficiency and Barlow disease) is the most frequent in Western countries [63]. According to the Carpentier functional classification, degenerative MV exhibits a type II dysfunction, with excessive leaflets motion, dominated by degenerative prolapse (see Figure 3). The first-choice imaging technique is echocardiography through the evaluation of qualitative, semi-quantitative, and quantitative parameters. If echocardiography is not sufficient for the quantification of the severity of valve insufficiency, cardiac magnetic resonance (CMR) is a valid alternative, especially in PMR, for the greater accuracy in the assessment of volumes. In patients with discordant symptoms and an MR grade at rest, exercise echocardiography is indicated to evaluate changes in the MR volume [64,65].

If MR develops acutely, urgent surgery is the only treatment. In cases of papillary muscle rupture, valve replacement is required. In chronic MR, the natural history of severe degenerative MR is poor. However, its correction at the right time interval is associated with a life expectancy similar to that of the normal population [66]. The surgical approach is recommended in patients with symptomatic severe PMR, which is judged to be operable by an HT. Additional triggers for considering surgery even in asymptomatic patients are: LVEF ≤60%, LVES diameter ≥40 mm, left atrium (LA) volume ≥60 mL/m^2^, systolic pulmonary arterial pressure >50 mmHg, and atrial fibrillation (AF) [1,67]. The surgical gold-standard treatment is MV repair, as it is associated with better survival compared to MV replacement. Transcatheter MV repair or valvular implantation for severe PMR are safe alternatives for patients with contraindications for surgery or a high operative risk [68,69,70,71]. Nowadays, despite the known poor prognosis of MR if it is not treated, MV surgery is performed only in a small percentage of patients, even in the highly repairable subset of degenerative MR (68,69). The association of PMR with sudden cardiac death (SCD) and ventricular arrhythmias (VA) remains controversial [72]. In addition to the aforementioned prognostic predictors, the presence of mitral annular disjunction (MAD) is to be counted (Figure 4). MAD consists in a disinsertion of the normal mitral annular structure, comprising the atrial–valvular–ventricular junction, with remaining posterior leaflet attachment on the atrial wall [73]. This confers an increased risk of ventricular arrythmias and SCD [74]. MAD usually involves P1 and P3 scallops of the posterior mitral leaflet [75] and occurs in about 30% of generally young patients with an MV prolapse diagnosis. Trans-thoracic echocardiography is the initial approach to detecting MAD, with 65% sensitivity and 96% specificity. CMR provides a better assessment of MAD through the evaluation of leaflet excursion and the mitral annular plane (see Figure 3). In addition, CMR allows for the detection of tissue fibrosis through the evaluation of late gadolinium enhancement. Advanced myxomatous degeneration, denoted by marked leaflet redundancy and bi-leaflet MVP, is the strongest MAD-associated MVP feature, whereas MR severity is not [73]. However, within the first 10 years post-diagnosis, MAD was not linked to excess mortality, and although the patient should be reassured from the survival point of view, careful monitoring for arrhythmias is necessary for MAD [76]. Secondary or functional mitral valve regurgitation (SMR) is a result of multifactorial left atrial and ventricular dysfunction and remodeling. It occurs in 11% to 59% of patients after acute myocardial infarction and is present in >50% of patients with dilated cardiomyopathy. It is also found as a consequence of LA enlargement and mitral annular dilatation in patients with longstanding AF or heart failure (HF) with preserved EF. The echocardiographic criteria of severity are the same as those of PMR, but even a milder degree contributes to a bad prognosis; if 2D echocardiography is not conclusive because of the asymmetrical shape of the regurgitant orifice, or in low-flow-state conditions, the use of 3D echocardiography, CMR, and exercise echocardiography may help to assess the entity of regurgitation. The staging of SMR is based upon symptoms, valve anatomy, and valve hemodynamics. It is mandatory to determine the cause of LV dysfunction; in fact, despite the prognosis being poor for both ischemic and non-ischemic SMR, documenting ischemic MR or large areas of the hibernating viable myocardium offers the possibility of revascularization and a potential improvement in LV function [1]. SMR is an independent driver of prognosis in patients with the intermediate HF phenotype but not in those with advanced HF [77]. For this reason, the follow-up of patients should include at least annual history and physical examination. The best therapy for chronic SMR is not clear because MR is only one component of the disease, and the restoration of mitral valve competence is not curative; therefore, the first therapeutic line is guideline-directed optimized medical therapy (OMT) for HF [78]. In a small and selected subset of patients with chronic severe SMR, LV systolic dysfunction, and persistent severe symptoms while on OMT, mitral transcatheter edge-to-edge repair is indicated to improve symptoms and prolong life [1], particularly if the patient has characteristics similar to those of the COAPT trial.

## 6. Tricuspid Regurgitation

The tricuspid valve (TV), once considered the forgotten valve, has been receiving recent increasing attention. Tricuspid regurgitation (TR) is a growing public health problem, as more than 4% of people >75 years old have a clinically relevant TR.

Despite the growing interest, global epidemiological data are lacking, with national screening studies revealing different prevalences: almost 4% of people >75 years old have a clinically relevant TR [79]. In the UK, 2.7% of older individuals were found to have moderate–severe TR, and in China, the prevalence was only 1.1% among patients of a similar age [2].

TR is an independent predictor of mortality and morbidity: the 3-year survival is about 58%, and the mortality increases with the worsening of the degree of TR. Unfortunately, more than 90% of patients with TR do not receive an operative and specific treatment for the regurgitation but are candidates only for general OMT, with variable results. On the other side, conventional surgery has an excessive mortality risk, with more than 10% in-hospital mortality. Therefore, given the poor prognosis of TR, efforts should be made to refer patients to the most effective treatment as soon as possible.

The rising interest in TR has led to a re-classification of its etiology, severity, and quantification methods. According to the Cleveland clinic, about 95% of TR are secondary, with left heart disease as the main cause (54,4%), followed by atrial functional (24.3%) and pulmonary disease (17%). The classification in primary TR, defined as a primary disruption of the structural integrity of the valve itself, and secondary TR, when there is an impaired valvular coaptation due to the distortion of the ventricular and/or atrial anatomy, has important implications not only for disease management but also for the prognosis [30,80,81]. Primary TR represents only 5% of the etiologies, mainly caused by endocarditis (47.2%) and degenerative/prolapse (18.3%) [80]. Patients with secondary TR had significantly worse survival than those with primary TR, likely due to their older age, comorbidities, and higher prevalence of heart disease [82]. Among secondary TR, the highest mortality is observed in TR secondary to pulmonary disease, reflecting the poor prognosis of the cor pulmonale, with severe RV dilation and dysfunction [80]. The new classification of TR goes beyond the simple distinction between primary and secondary, promoting the notion that “not all secondary TRs are the same” [83], separating them into a ventricular and an atrial form, and adding cardiac implantable electronic device (CIED)-induced TR. Due to the increasing number of device implantations, partly related to the aging of the population, the latter is expected to increase over time [1]. Secondary ventricular TR is characterized by marked leaflet tethering, systolic leaflet restriction, and RV dilatation and disfunction; secondary atrial TR is diagnosed by exclusion, in the absence of any leaflet abnormality, LV dysfunction, pulmonary hypertension, or CIED, and on the basis of a longstanding or permanent AF. This new classification also has prognostic implications; indeed, atrial TR has a rapid progression and a very poor outcome, and secondary ventricular TR occurs in advanced stages of cardiac diseases. Considering that CIED-induced TR shares primary and secondary TR features, it has been proposed as a third distinct category. Primary CIED-induced TR can be defined as an increase in TR severity of two grades during follow-up after CIED implantation in patients with documented interference of the device lead with the TV apparatus. Conversely, secondary CIED-induced TR is the consequence of the remodeling of the TV following the RV dilatation due to pacing/HF [84]. New guidelines suggest a multimodality imaging approach, integrating echocardiography, CT and CMR in order to overcome the intrinsic limitations of each technique. However, according to new guidelines, echocardiography is the first-choice imaging modality, and the severity classification of TR is based on qualitative, semi-quantitative, and quantitative echocardiographic parameters. Regardless of the imaging modality, the foundation of TR severity assessment is a thorough study of its anatomic and functional substrates [85]. To decide the correct therapeutic strategy, it is necessary to understand, in addition to the etiology of TR, the pathophysiological evolution. As for the functional TR, its pathophysiology consists of three phases: the first is characterized by an initial RV dilatation resulting in a tricuspid annulus dilatation without the development of pulmonary hypertension (PH), the second is characterized by a progressive RV and tricuspid annulus dilatation that results in a lack of leaflet coaptation, and the third is characterized by the progressive distortion of the RV, with an important tethering of leaflets and pulmonary hypertension [85]. Before any interventional approach, it is mandatory to assess the patient hemodynamic profile and, in particular, to evaluate whether, in addition to the increased pulmonary pressure, there has been an increase in the transpulmonary gradient, because, if this is the case, OMT is the only possible therapeutical approach. If pre-capillary or combined PH develops, the post-interventional physiological reverse remodeling, which usually slowly occurs after the reduction in TR, cannot happen, and there will be a progressive exhaustion of the RV and, inevitably, patient exitus. Hence, only identifying the underlying mechanism of regurgitation and integrating the knowledge on the etiology of the disease with the assessment of patient conditions will allow for choosing the correct therapeutical approach [86]. The three possible therapeutic solutions for TR are surgical TV repair or replacement, transcatheter treatment, and OMT. In view of the above considerations, i.e., the high mortality of conventional surgery and the bad prognosis with OMT, percutaneous treatments are slowly making their way. The current guidelines recommend transcatheter treatment only for patients with isolated, secondary TR, without severe RV/LV dysfunction or severe PH in the presence of symptoms, and for individuals that, according to the HT, are not appropriate for surgery. As a consequence, the window of transcatheter treatment is currently very small, but, hopefully, in view of the early success of this approach, things will change in the coming years. The percutaneous approach includes the most common leaflet approximation with devices such as TriClip or PASACAL, annuloplasty (through the Cardioband device [87] or Millipede device), orthotopic valve replacement (mainly with the Evoque prosthesis), and heterotopic valve replacement, for patients with a difficult anatomy or with advanced dysfunction, with prostheses such as Tricento or Tric-Valve implanted in the caval vein [88].

We summarized the epidemiologic and diagnostic features of VHD in Table 1 and the implications for follow-up and management in Table 2. In Figure 5 we show the prevalence of non-rheumatic VHD.

## 7. Multiple VHD

Multivalvular heart (MHD) disease is defined by the presence of regurgitant and/or stenotic lesions in two or more cardiac valves; despite its high prevalence, there is a considerable lack of data in the current literature in terms of evidence-based recommendations regarding its clinical management.

Thanks to the improvement of living conditions, nutrition, and access to medical care (especially, the spread of penicillin), the predominant pathogenesis of MHD has consistently changed: at the beginning of the 21^st^ century, RHD was the predominant etiology [89]; over the years, according to the EuroObservational VHD II survey, the incidence of RHD has dramatically declined in developed countries, and, as a consequence of the aging of the population, the degenerative etiology has become prevalent, being about 60% versus 20.5% of RHD [90].

Most frequently, MHD consists of the copresence of left-sided valve disease with TR [90]. The second most common association is severe AS with moderate/severe MR, reaching up to 20% of patients undergoing aortic valve replacement [91].

The combination of AS with functional-mitral regurgitation (FMR) is frequent. In most cases, FMR is mild, but this combination has intrinsic pathophysiological complexity [91]. In the presence of AS, with relatively small ventricles with concentric hypertrophy, [92] and high end-diastolic intraventricular pressure, even a modest MR can reduce the stroke volume and increase the LA pressure, promoting LA dysfunction, with the consequent activation of neurohormonal signals and LV dysfunction [93]. The female sex, a low body mass index, and elevated baseline right ventricular (RV) systolic pressure are predictors of persistence or worsening baseline MR in the presence of AS [94]. Patients with concomitant AS and FMR have a higher risk of HF or death, particularly during the medical follow-up [95]. Despite current guidelines trying to expand the criteria for patients who would benefit from a valvular replacement and trying to identify new variables that can help risk stratification, there is not yet a specific indication for concomitant FMR [81]. In the past, most patients with concomitant AS and MR would be treated by double-valve surgery, with a predicted in-hospital mortality of 8% [96]. Now, this approach should be revised: indeed, one of the main characteristics of FMR, in the presence of AS, is a great reduction after SAVR/TAVR, without the need for double surgery. The results of the PARTNER trial showed that moderate/severe MR improved in 69.4% of SAVR patients and in 57.7% of TAVR patients [97]. Therefore, cardiologists should focus on both AS and FMR quantification, considering them as a whole, in order to improve the management of the individual patient, especially during the long period of medical follow-up. Thus, FMR should be investigated through functional capacity evaluation [98].

However, regardless of the involved valves, patients with severe multiple VHD are more symptomatic, have a higher incidence of HF at 6 months, and have a worse prognosis compared with those affected by monovalvular disease [90].

## 8. Heart Valve Centers

The increasing number of patients with VHD and the wide range of therapeutic options now available demand a dedicated management and a standardization of procedures [1] in order to refer patients to a surgeon or interventional cardiologist at the appropriate timing before the development of changes in the LV or major adverse clinical events. Frequently, patients with VHD are asymptomatic at presentation, and the disease progression may not be recognized by physicians without specialist competencies. Typical is the case of AS, whereby about one-third of patients are referred for intervention either too early or too late [16].

The central role of the VHD specialist has been emphasized by both American [30] and European [1] guidelines. This professional figure should achieve specific skills and appropriate expertise that are not included in the classic training courses for interventional cardiologists, cardiovascular surgeons, echocardiographers, or HF specialists.

In order to ensure the appropriate management of inpatients and outpatients or in the follow-up after the procedure, several professional profiles are involved in the valve center: clinical and interventional cardiologists, cardiac surgeons, imaging specialists with expertise in interventional imaging [99], cardiovascular anesthesiologists, and, if necessary, HF specialists for the evaluation of the patient with secondary MR and TR, electrophysiologists, and specialists in infectious disease and/or medical microbiology. Other team members include cardiovascular nurses and sonographers, elderly care physicians, pulmonologists, nephrologists, microbiology and infectious disease specialists, neurologists, and psychiatrists. All members of the multidisciplinary HT need to be involved in continuing education appropriate to their roles [100].

The multidisciplinary approach is recommended for all types of VDH and has been formally endorsed by several Scientific Societies [101]. However, it is time-consuming, and it might be difficult to gather the HT participants. Thus, it is important to promote HT meetings by scheduling them in the agendas of specialists. There is therefore a need to create networks to improve communication both with outpatients and with all the doctors involved in the treatment, also resorting to teleconferences but, more immediately, using digital communication systems based on mobile phones or, where possible, consulting the patient’s passport. The creation of the networks would facilitate transfers from a district hospital to a valve center.

The procedures that should be available at heart valve centers are valve replacement in all four sites: mitral and TV repair; AF ablation; TAVR; and surgery for the aortic root and ascending aorta.

The relationship between the case volume and outcome of surgery and transcatheter interventions is complex, and the precise number of procedures per individual operator or hospital required to provide high-quality care remains controversial, as inequalities exist between high- and middle-income countries [102]. Furthermore, these data come from retrospective registries, suggesting that the thresholds may be too low in current practice [100]. For this reason, the ability to obtain good results is more important than mandating volume targets.

Robust internal audits [103] with regular outcome assessment or morbidity and mortality meetings and the reporting of near misses are essential. The center should report at least 30-day, 1-year, and 5-year mortalities [104]. Commonly used risk scores (e.g., Society of Thoracic Surgery (STS) score or Euroscore II), including frailty scores for transcatheter valve procedures, should be available to interpret outcome data at the level of individual patient risk, despite their limitations [104].

All valve centers require consistent access to high-quality echocardiography, including 3D, and an echocardiographer with expertise in VHD [105]. Other modalities such as CMR and CT provide additional information and help in the risk assessment of some patients, particularly if the echocardiographic images are suboptimal. Positron emission tomography (PET) should also be available, since the 2015 modified criteria include PET evidence as a major criterion in the diagnosis of prosthetic valve endocarditis [106].

The members of the multidisciplinary team of the VHD center should collaborate to develop individualized care plans, continually working together to provide patients with innovative and effective cardiovascular care.

## 9. Conclusions

VHD is a leading cause of cardiovascular morbidity and mortality, with substantial regional differences. Population aging, the availability of imaging techniques, accessibility to diagnosis and treatment, migration flows, the improvement in valve surgery, and the advent of transcatheter procedures are the principal factors responsible for changes in the epidemiology of VHD. These must be well known to the physician and healthcare organizer/legislator. Improvements in the prevention, detection, and treatment of VHD are necessary in order to reduce the global healthcare burden.

## Figures and Tables

**Figure 1 jcm-12-02178-f001:**
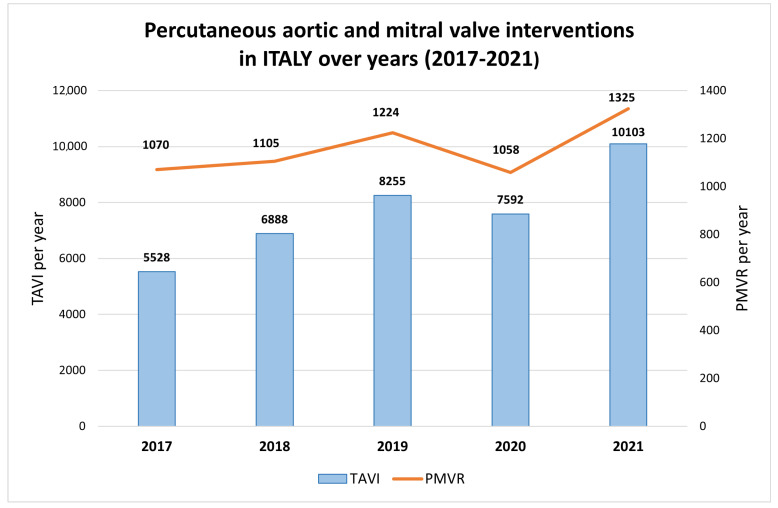
Percutaneous aortic and mitral valve interventions in Italy over the years (2017–2021). Data from the GISE (Gruppo Italiano Studi Emodinamici) Registry. TAVI: Transcatheter aortic valve implantation. PMVR: Percutaneous mitral valve repair.

**Figure 2 jcm-12-02178-f002:**
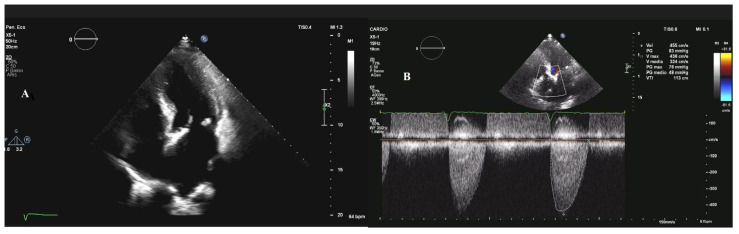
**Severe aortic stenosis.** Panel (**A**) shows an apical four-chamber view with the hypertrophic left ventricle; panel (**B**) shows the peak aortic velocity (4.3 m/s) and the median transvalvular gradient (48 mmHg) evaluated by continuous-wave Doppler ultrasound.

**Figure 3 jcm-12-02178-f003:**
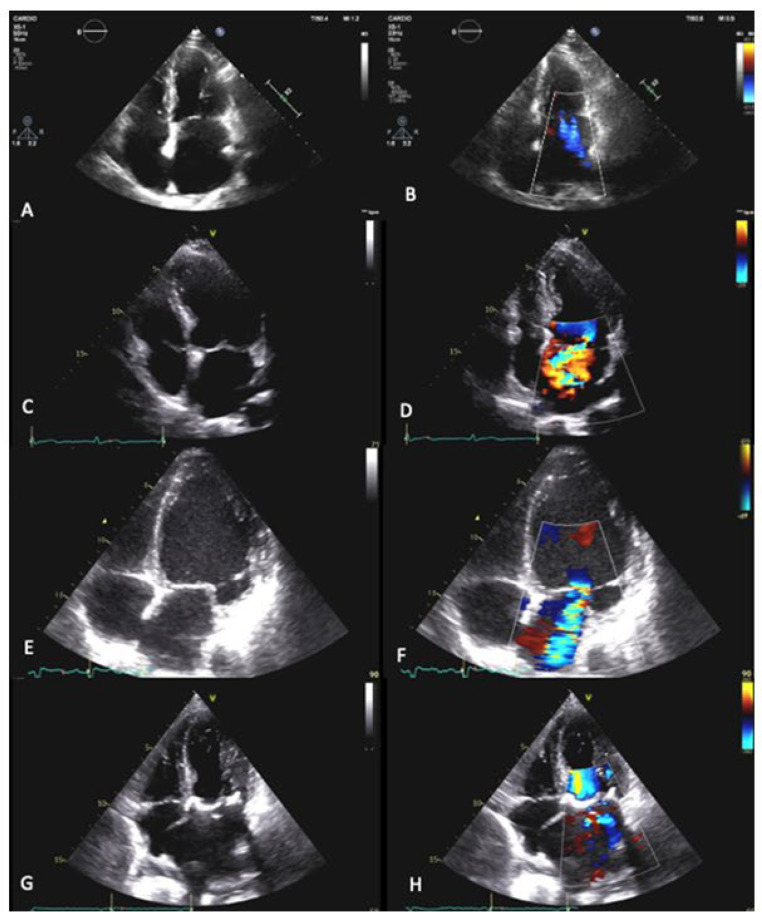
**Carpentier classification of mitral regurgitation (MR)**. Apical four-chamber view, on the left, and Color Doppler ultrasound evaluation, on the right. (**A**,**B**) Type I: normal leaflet motion; annular dilatation. (**C**,**D**) Type II: excessive leaflet motion (mitral prolapse or flail). Type III: restricted leaflet motion; (**E**,**F**) IIIa: restricted opening during systole and diastole (rheumatic disease); (**G**,**H**) IIIb: restricted closure during systole (ischemic MR).

**Figure 4 jcm-12-02178-f004:**
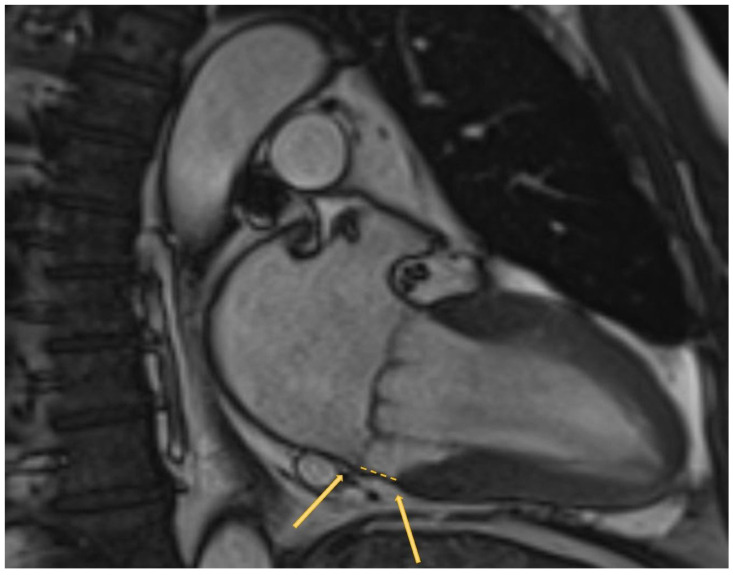
**Mitral annular disjunction.** Mitral atrioventricular disjunction (MAD). Cardiac magnetic resonance (CMR) two-chamber long-axis: the yellow arrows indicate the site of atrioventricular junction used to assess the MAD (from the top edge of the ventricular wall to the hinge of the leaflet from the left atrial wall).

**Figure 5 jcm-12-02178-f005:**
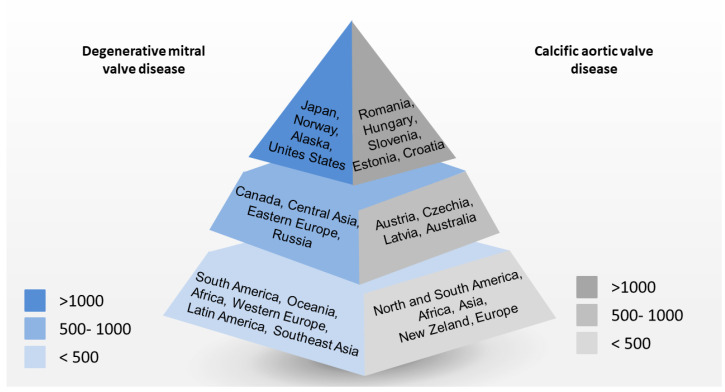
Prevalence of non-rheumatic Valve Heart Disease per 100,000 people.

**Table 1 jcm-12-02178-t001:** Characteristics of valvular heart disease.

Valvulopathy	Etiology	Anatomy	Prevalence	Risk Factors
Aortic Stenosis	Degenerative (81.9%), rheumatic (11.2%), Congenital (5.6%), post-endocarditis (1.3%)	CAVD: Aspecific Bicuspid valve: fusion between right and left leaflets is the most common Rheumatic: commissural fusion	3.4% in the elderly	Aging, hypertension, diabetes, chronic kidney disease
Aortic regurgitation	Congenital bicuspid AV, calcific degeneration, rheumatic disease (in developing countries), myxomatous degeneration	Type I: aortic root dilatation + coaptation defect Type II: leaflet prolapse Type III: leaflets degeneration and retraction	1.6% in UK elders aged >65 years old; 4.9% of participants in the US Framingham study	Hypertension, aging, Marfan syndrome, Turner’s syndrome
Mitral Regurgitation	Primary: endocarditis, degenerative, papillary muscle rupture. Secondary: ischemic, dilated cardiomyopathy, atrial enlargement caused by AF	Type I: normal leaflet motion Type II: excessive leaflet motion Type III: restricted leaflet motion IIIa: leaflet motion restricted in both systole and diastole. IIIb: leaflet motion restricted in diastole	up to 10% of the general population; mitral valve prolapse: 3% of the general population.	Myxomatous degeneration (in the younger population), fibroelastic deficiency disease (in the elderly), LV dysfunction, LA dilatation
Tricuspid Regurgitation	Primary: endocarditis, degenerative, prosthetic valve failure, implantable device-related Secondary: left heart disease, atrial functional, pulmonary disease, right heart disease.	-annular dilatation -leaflet tethering, leaflet restriction	4%	Aging, atrial arrythmias, pulmonary hypertension, RV dysfunction, lead in the right ventricle.

AV: Aortic Valve. CAVD: Calcific aortic valve disease. LA: Left Atrium. LV: Left ventricle. RV: Right Ventricle. UK: United Kingdom. US: United States.

**Table 2 jcm-12-02178-t002:** Management of valvular heart disease.

Valve Disease	Follow-Up	Medical Therapy	Intervention
Aortic Stenosis	Mild: every 3–5 y Moderate: every 1–2 y Severe: every 6–12 mo	N.A.	Severe AS in symptomatic patients or asymptomatic patients with LV systolic dysfunction or a positive stress test − TAVR: STS-PROM/EuroSCORE II >8%, Age ≥75 years, contraindications for surgery − SAVR: STS-PROM/EuroSCORE II <4%, Age <75, unfavorable anatomy for transfemoral TAVR − balloon aortic valvotomy: bridge to SAVR or TAVR in hemodynamically unstable patients
Aortic Regurgitation	Mild: every 3–5 y Moderate: every 1–2 y Severe: every 6–12 mo	− When surgery is not indicated: control hypertension − Losartan + atenolol decreases the rate of progression in Marfan syndrome	− symptomatic patients regardless of LVEF − asymptomatic patients with LVEF <50% or LVESD <25 mm/m2 BSA − necessity of CABG or surgery for ascending aorta or another valve; the intervention is indicated in severe symptomatic or asymptomatic AR
Mitral Regurgitation	Mild: every 3–5 y Moderate: every 1–2 y Severe: every 6–12 mo	− When surgery is not indicated − Medical therapy for systolic dysfunction (beta-blockers, ACE-I, and, if needed, aldosterone antagonists)	− acute papillary muscle rupture Primary MR: − symptomatic patients regardless of LVEF − asymptomatic patients with LVEF ≤60%, LVES ≥40 mm, left atrium (LA) volume ≥60 mL/m^2^, systolic pulmonary arterial pressure >50 mmHg, and atrial fibrillation.
Tricuspid Regurgitation	Non-specific follow-up period.	− When surgery is not indicated − Diuretics, anti-arrhythmic drugs, HF therapy	− patients undergoing left-sided valve surgery: − with severe functional TR; − symptomatic or in the presence of progressive right ventricular dilatation or dysfunction after previous left-sided surgery − patients with mild or moderate functional TR with a dilated tricuspid annulus undergoing left-sided valve surgery

AS: Aortic Stenosis. BSA: Body Surface Area. CABG: Coronary artery bypass graft. HF: Heart Failure. LA: Left Atrium. LVEF: Left Ventricular Ejection Fraction. LVSED: Left ventricular end-systolic diameter. SAVR: Surgical aortic valve replacement. TAVR: Transcatheter aortic valve replacement. TR: Tricuspid regurgitation. N.A: not available.

## Data Availability

Not applicable.

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
