# Peer review of "The Global Burden of Valvular Heart Disease: From Clinical Epidemiology to Management"

_jcm, 2023, doi:10.3390/jcm12062178_

Round 1
Reviewer 1 Report
Esteemed,
I had the pleasure and honor to review the paper: „THE GLOBAL BURDEN OF VALVULAR HEART DISEASE: FROM CLINICAL EPIDEMIOLOGY TO MANAGEMENT“, by Santangelo and associates. This narrative review focuses on a very important issue regarding the permanently changing epidemiology of primary heart valve diseases (VHD).
The entire manuscript is well-composed, covering the most relevant aspects of primary HVD epidemiology, yet, there is one suggestion:
· In section „B“of the Introduction (lines 62-128) I would add a paragraph about the factors that have resulted in the changing face of INFECTIVE ENDOCARDITIS. It would be interesting and important for the paper of this profile to comment on the shift from Streptococcus sp. to more virulent Staphylococcus sp. considering inadequate antibiotic use and consequent increase in antibiotic resistance.
Author Response
Author: Thanks to the Reviewer for the relevant observation. As suggested, we have added the following paragraph to the introduction section:
“-The increase in the predisposing factors of infective endocarditis. The last 20 years have witnessed an increase in all the main factors predisposing to infective endocarditis, such as the aging population, increased use of intracardiac, vascular and valvular devices (e.g. pacemakers, defibrillators, biological and mechanical valve prostheses), epidemic levels of opioid addiction and associated injection drug use. Furthermore, the inadequate and unholy antibiotic use with consequent increase in antibiotic resistance has led to the dangerous shift from Streptococcus to more virulent microorganisms, such as Staphylococcus and Enterococcus, as more common causative organisms of infective endocarditis”.
Reviewer 2 Report
The authors here aimed to summarize information on valvular heart disease in a review article, with a special focus on aspects of epidemiology and clinical management. They conclude that there is a growing epidemiologic need of treatment for patients with valvular heart disease and that there are more and distinctly different strategies available for treatment - and thus specialized "heart valve centers" with high patient volumes and externsive expertise should be trusted with diagnostics and treatment decisions in this complex field.
While the topic is timely, I have some concerns that should be adressed:
Major points:
1. The manuscript lacks Figures to support the lengthy text passages: Figure 1 is good, but the other figures (2+3+4) seem to be chosen at random and do not convey the message appropriately. Data on valvular heart disease epidemiology should be easy to display in Figures and/or Tables, instead of resorting to text passages and references.
2. The manuscript completely lacks a conclusion. Please revise.
3. The title promises "global" data on the burden of valvular heart disease, however predominant perspectives in the article are European/Western. Please either contribute more data from the world or change the title.
4. Figure 5 is inappropriate as a graphical abstract of a heart valve center. Please revise.
Minor points:
English language in some parts needs editing.
